# A Review of the Phytochemistry and Pharmacology of the Fruit of *Siraitia grosvenorii* (Swingle): A Traditional Chinese Medicinal Food

**DOI:** 10.3390/molecules27196618

**Published:** 2022-10-05

**Authors:** Juanjiang Wu, Yuqing Jian, Huizhen Wang, Huaxue Huang, Liming Gong, Genggui Liu, Yupei Yang, Wei Wang

**Affiliations:** 1TCM and Ethnomedicine Innovation & Development International Laboratory, Innovative Materia Medica Research Institute, School of Pharmacy, Hunan University of Chinese Medicine, Changsha 410208, China; 2School of Chinese Medicine, Macau University of Science and Technology, Macau 999078, China; 3Hunan Huacheng Biotech, Inc., High-Tech Zone, Changsha 410205, China

**Keywords:** *Siraitia grosvenorii*, Cucurbitaceae, chemical compounds, pharmacological effects, traditional Chinese medicine, mogrosides, *Luohanguo*

## Abstract

*Siraitia grosvenorii* (Swingle) C. Jeffrey ex Lu et Z. Y. Zhang is a unique economic and medicinal plant of Cucurbitaceae in Southern China. For hundreds of years, Chinese people have used the fruit of *S. grosvenorii* as an excellent natural sweetener and traditional medicine for lung congestion, sore throat, and constipation. It is one of the first species in China to be classified as a medicinal food homology, which has received considerable attention as a natural product with high development potential. Various natural products, such as triterpenoids, flavonoids, amino acids, and lignans, have been released from this plant by previous phytochemical studies. Phar- macological research of the fruits of *S. grosvenorii* has attracted extensive attention, and an increasing number of extracts and compounds have been demonstrated to have antitussive, expectorant, antiasthmatic, antioxidant, hypoglycemic, immunologic, hepatoprotective, antibacte- rial, and other activities. In this review, based on a large number of previous studies, we summarized the related research progress of the chemical components and pharmacological effects of *S. grosvenorii*, which provides theoretical support for further investigation of its biological functions and potential clinical applications.

## 1. Introduction

The genus *Siraitia* contains four species in China, *S. grosvenorii*, *S. borneensis*, *S. borneensis* var. yunnanensis, *S. taiwaniana*, *S. siamensis* [1], which are all native to the southern parts of China, mainly in Guangxi and Hunan Provinces. Among them, the most famous species is *Siraitia grosvenorii* (Swingle) C. Jeffrey ex Lu et Z. Y. Zhang, which is a perennial herbaceous vine. The formal Chinese name of *S. grosvenorii* is Luohanguo (罗汉果), and locally known as *monk fruit*, *lahanguo*, *jiakugua*, or *guangguomubie* [1]. The shape of fruits is ovoid, oval or spherical, the surface of fruits is brown, yellowish brown or greenish brown, with yellow soft hairs brown, tawny or greenish-brown, with yellow fuzz, which is indeed light and lightweight, the texture is crisp and tender, the taste is sweet. As a healthy food, the fruits of *S. grosvenorii* have pharmacological activities including immune-enhancing, liver-protecting, hypoglycemic, anti-tumor, and antioxidant [2,3,4,5,6]. As a traditional Chinese medicine, the fruits of *S. grosvenorii* could be used to treat cough and asthma, sore throat, constipation, and so on [7]. Previously research hadshown that *S. grosvenori**i* contains triterpenoids, flavonoids, lignans, vitamins, proteins, saccharides, and volatile oil [8]. Especially, the plants’ ripe fruits could be used as supplement and sweetener in sugar-free healthy foods and drinks for commercial use, due to their high content of natural low-calorie glycosides [9]. 

Mogrosides, glycosides responsible for sweetness, were isolated from the fruits of *S. grosvenorii*, which are considered to base be the main active ingredients in the sweetness and biological function of this plant. Existing studies have shown that the extracts and individual compounds of *S. grosvenorii* were non-toxic (Figure 1). A range of related products has been approved as dietary supplements in Japan, the United States, New Zealand, and Australia. Hunan Huacheng Biotech., Inc., a Chinese exporter of mogroside sweeteners, is the world’s leading expert in *S. grosvenorii* extraction, providing sugar reduction solutions. For the sacred mission of bringing the pure, natural, zero calorie and healthy sweet to everyone, they have carried out in-depth cooperation with well-known domestic food and natural product research institutions and established established a complete quality management system with state-of-the-art extraction, separation, and purification technology. In 2012, the extract of *S. grosvenorii* was included on the priority list for the 44th session of the Codex Committee on Food Additives, Joint FAO/WHO Food Standards Programme held in Hangzhou, China. In past years, Hunan Huacheng Biotech, Inc. submitted a GRAS notice to the FDA. The substances which were the subject of the notification were a clarified concentrate derived from the fruits of *S. grosvenorii* and trademarked and sold as H2-Luo^TM^ brand *luohanguo* concentrate. The primary components of the concentrate were mogrosides, with mogroside V constituting more than 25% of the product. Hunan Huacheng Biotech, Inc. received a “no questions at this time” response from the FDA. At the same time, H2-Luo^TM^ Monk Fruit Blend Sweetener has been widely available on the market, which it has the same sweetness and usage as sugar, and could be used in candy, drinks, solid drink, table sugar, health products, and dairy products. In particular, the pharmaceutical preparations containing this plant have become the mainstay of clinical use in the treatment of throat disorders and lung diseases. This review summarizes the chemical constituents and pharmacological activities of *S. grosvenorii*, which expect to providing a reference for the research of the therapeutic basis and development of new drugs, while also providing value for in-depth study and utilization of Cucurbitaceae.

## 2. Traditional Uses

The fruits of *S. grosvenorii* with a long history displays a wide range of biological activities, especially their therapeutic effects on throat disorders including tuberculosis and whooping cough. Moreover, the fruits of *S. grosvenorii*, which were have been used medicinally for over300 years since recorded, have a high nutritional value and were one of the first valuable Chinese herbs to be included in medicine and food. *S. grosvenorii* was first recorded in the Chinese medical classic “Xiu Ren Xian Zhi” (Qing dynasty). *S. grosvenorii* was described as a herbal medicine with a diuretic effect, and function for clearing away heat and treating coughs. Based on “Chong Xiu Lin Gui Xian Zhi” (Qing dynasty), *S. grosvenorii* was described as a therapy for coughs caused by tuberculosis. In traditional Chinese medicine (TCM) theory, *S. grosvenorii*, widely used in TCM with few side effects, is sweet, slightly cold, and goes to the lung and large intestine meridian. The Chinese Pharmacopoeia recommends a dose of 9-15 g for *S. grosvenorii*, which has the properties of clearing away heat and moistening the lungs, relieving sore throat to restore voice, and loosening the bowel to relieve constipation. Clinically, the fruits of *S. grosvenorii* as pharmaceutical preparations had similar effects [10].

## 3. Chemical Composition

Previously, several different classes of secondary metabolites were isolated from the fruits of *S. grosvenorii*, while some others were isolated from its roots and leaves. So far, nearly 100 compounds have been isolated and identified from the title plant, including triterpenoids, flavonoids, and lignans, etc. [11,12]. Cucurbitanes, a class of tetracyclic triterpenoids, are the most dominant structural types in the plant, which they could be divided into general cucurbitanes and nor-cucurbitanes. The main structural differences between these compounds are the number and position of sugar chains attached to the same cucurbitane type skeleton, including monosaccharides, and disaccharides, and trisaccharides, and sugar chains usually connected to C-2, C-4, or C-6 positions on the skeleton (Figure 2).

### 3.1. Triterpenoids and Glycosides

To date, a total of 71 triterpenoids have been isolated from the fruits of *S. grosvenorii*, including 56 cucurbitanes (Table 1), 9 nor-cucurbitanes, 1 lanostane, and 5 oleananes (Figure 3). Cucurbitadienol is a basic precursor of cucurbitane tetracyclic triterpenoids isolated from a variety plants of Cucurbitaceae. For example, secondary metabolites generated from the *S. grosvenorii* plant, such as mogrosides, which are the main active components of the fruits of *S. grosvenorii*. Most of these are sweet, accounting for about 1.19% in fresh fruits [13] and 3.82% in dried fruit powder [14]. Research on *S. grosvenorii* began in 1974, and American Lee Chihong firstly extracted crude mogrosides [15]. In 1977, Japanese researcher Takemoto Tsunematsu began studies to isolated pure mogrosides and deduced their corresponding chemical structures [16]. In 1983, mogrosides IV (**8**), V (**16**), and VI (**18**) were successfully isolated from the fruits of *S. grosvenorii*. Then, more than 40 analogues were obtained in succession. All these compounds have mogrol, [10-cucurbit-5-ene-3,11,24*R*,25-tetraol], attached to different numbers of glucose units. Based on the variation of mogrol at positions C-7, 11, and 25, mogrosides could be roughly classified into five categories. The main differences are whether there is a carbonyl group at C-7, a conformational change with a hydroxyl group attached to C-11, and whether there is a methyl group at C-25. Mogrosides are distinguished by the hydroxyl or carbonyl groups of C-11. While most mogrosides have an 11α-OH and only one has been identified as 11β-OH which is defined as 11-epimogroside V (**31**). Some mogrosides also have a carbonyl group at C-7 (**26**–**29**) or lose a hydroxyl group at C-25 (**53**–**56**), but no relationship has been found between their structures and pharmacological activities. 20-hydroxy-11-oxo-mogroside IA_1_ (**47**) was obtained from the unripe fruits of *S. grosvenorii* [17]. As shown in the research, the structure-taste relationship of the 3*β*-hydroxy-cucurbit-5-ene derivative glycosides depends on the number of glucose units, the oxygen function at the C-11 position of the aglycone moiety, the location of the glucose group units, and the hydroxylation of the side chain [18,19,20]. Mogroside IIE (**12**), mogroside IIIE (**13**) and mogroside III (**15**) are tasteless, because the basic structural requirement of at least three sugar units is not met. 11*α*-hydroxyl glycosides taste sweet, such as compound mogroside IVA (**9**) and mogroside IVE (**14**). In contrast, the 11*β*-hydroxyl glycosides are tasteless, and the 11-oxide compounds and their dehydrogenated derivatives taste bitter. It is noteworthy that the relationship between the distribution of glucose units and sweetness. Interestingly, the more sugar units, the higher the corresponding sweetness. Siamenoside I (**21**) has 5 glycoside units and is the sweetest triterpenoid glycoside compound isolated so far, which has a similar sweetness to mogroside V (**16**). Mogroside V (**16**) is the main compound of the fruits of *S. grosvenorii* with an average content over of 0.5% [21]. Mogroside V (**16)** and siamenoside I (**21**) at 1/10000 concentration were 425 and 563 times sweeter than 5% sucrose, respectively [19]. However, the number of glycoside units of mogroside IVA (**9**) and mogroside IVE (**14**) is the same as siamenoside I (**21**), but the sweetness is lower than siamenoside I (**21**). The hydroxylation of side chains could cause taste changes. For example, the bitter 11-oxo-mogrosides are sweetened by hydroxylation of the side-chain double bond with osmium tetroxide. Moreover, the mogrosides content in fruits changes during the growth and development stage. Mogrosides appeared five days after pollination, and mogroside IIE (**12**) is the main constituent in young fruit for the first 30 days; Mogroside III (**15**) began to appear after 30 days, it reached its maximum content in 55 days, then formed mogroside IV (**8**), mogroside IVA (**9**) and mogroside IVE (**14**). The content of mogroside IV (**8**) peaked at 70 days; mogroside V (**16**) was produced after 70 days and developed into the main sweet peaked about 85 days after pollination, and then the fruit began to mature [22]. Therefore, the best time to harvest the fruit should be 90 days after pollination (Figure 3). Besides, the fruits have a bitter taste for the first 50 days. As the growth cycle increases, the taste of fruit gradually changes from bitter to sweet [23,24]. Excepts for mogrosides, other cucurbitanes and oleanolanes are also present in this plant. For example, cucurbitanes are 5*α*,6*α*-epoxymogroside IE_1_ (**32**), 25-dehydroxy-24-oxo-mogrol (**53**), 3-hydroxy-25-dehydroxy-24-oxo-mogrol (**54**), bryogenin (**55**) and 10*α*-cucurbitadienol (**56**) and oleanolanes are isomultiflorenol (**67**), *β*-amyrin (**68**), karounidiol 3-benzoate (**69**), karounidiol dibenzoate (**70**), 5-dehydrokarounidiol dibenzoate (**71**). These compounds were identified from different parts of *S. grosvenorii* [25,26]. Furthermore, mogroseter (**66**) is the first cycloartane to be discovered from this plant so far, which was reported in 1992 from the leaves of *S. grosvenorii* [27,28]. Its structure is characterized by a lanostane-type skeleton although the methyl group at C-19 is dehydrogenated with C-9 to form a cyclopropane. Under acidic conditions, it could undergo a ring-opening reaction to produce a cucurbitane-type compound. Recently, researchers have gained new insights into siraitic acids, the methyl group at C-26 is oxidized to a carboxyl group. The forms of siraitic acid present in *S. grosvenorii* are divided into two kinds: one is in the form of free acid, such as siraitic acid A-F (**57–58**, **60–63**) [29,30,31]; the other exists in the form of glycoside combined with sugar group, such as siraitic acid IIA (**65**), IIB (**59**) and IIC (**64**) [32]. They were isolated from the roots of *S. grosvenorii.* Siraitic acids are plant-specific cucurbitane compounds known as nor-triterpene acids, whereas mogrosides are triterpenoids glycosides. Nor-cucurbitacin triterpenoids, are formed by the reduction of one or several carbons on the 30-carbon skeleton of cucurbitacin-type triterpenoids. According to the position of the vacant carbon atom, nor-cucurbitacin triterpenoids are divided into two types, including 29-nor and 19,29-nor [33]. **57** to **60** are of the 29-nor type, and the main difference between them and cucurbitanes is the loss of a methyl group at C-4, while the A-ring is also aromatic, and they belong to the monomeric cucurbitanes. **61** to **65** are 19,29-nor type, and they are based on the cucurbitane triterpene skeleton with a methyl deletion at C-4 in the A-ring. A double bond between C-4 and C-5 and a methyl deletion at C-9 in the B-ring, thus they are called dinor-cucurbitanes. The corresponding structures of these main cucurbitanes components isolated from *S. grosvenorii* are listed in Table 1 and shown in Figure 4.

### 3.2. Flavonoids

Besides triterpenoids, diverse flavonoids are present in *S. grosvenorii* [63,64]. Surprisingly, flavonoids reported in this plant were very few, and only eight (**72**–**79**) were mentioned in previous literature. Flavonoids are compounds with a 2-phenylchromanone (flavone) structure that are widely occurring in nature and are the most abundant polyphenols in the human diet, accounting for about 2/3 of the total intake. The main flavonoid types in this plant are general flavones and flavonols, which possess the quercetin or kaempferol skeleton with sugar groups. In 1994, grosvenorine (**74**) and kaempferitrin (**75**) were first isolated from the fruit of *S. grosvenorii* [63]. Grosvenorine (**74**) is the main flavonoid glycoside in *S. grosvenorii*. The flavonoid glycosides of leaves were separated and purified by solvent extraction, column chromatography, and HPLC purification [65]. From this, two flavonoid glycosides were discovered, kaempferitrin (**75**) and quercetin 3-O-β-D-glucopyranosyl 7-O-α-L-rhamnopyranoside (**76**). The flowers contained kaempferol (**72**), kaempferol 7-O-α-L-rhamnopyranoside (**73**), grosvenorine (**74**), 7-methoxy-kaempferol 3-O-α-L-rhamnopyranoside (**77**) and 7-methoxy-kaempferol 3-O-β-D-glucopyranoside (**78**) [27,66,67]; afzelin (**79**) was isolated from the crude extracts [68], the corresponding structures of flavonoids isolated from *S. grosvenorii* are listed in Table 2 and relatively shown in Figure 5.

### 3.3. Miscellaneous Componds

The main chemical constituents in *S. grosvenorii* are triterpenoids, flavonoids and their glycosides, while other constituents are less frequently reported. So far, three phenolic acids, two anthraquinones, three alkaloids, three fatty acids, and several other compounds have been isolated and identified, the corresponding structures isolated from *S. grosvenorii* are listed in Table 3 and shown in Figure 6. 

### 3.4. Polysaccharides

The polysaccharides in *S. grosvenorii* are mainly composed of mannose, arabinose, and xylose, with high levels of glucose [76]. Recently, a novel polysaccharide SGP-1-1 was obtained. It is an acidic polysaccharide and composed of arabinose (Ara), ribose (Rib), galacturonic acid (GalAc), galactose (Gal), mannose (Man), and glucose (Glc) at the molar ratio of 1.00:1.72: 2.24: 3.64: 3.89: 22.77 [77].

### 3.5. Amino Acids & Proteins

The protein content of *S. grosvenorii* ranged from 8.67 to 13.35% in fresh fruit and 7.1 to 7.8% in dried fruit. In hydrolysate, 18 amino acids were complete. These amino acids included eight amino acids that are essential for the human body. The highest contents were glutamic acid and aspartic acid [78], showing that *S. grosvenorii* has a certain nutritional value.

## 4. Phytochemical Analysis 

Analytical techniques provide reliable and efficient methods for the phytochemical analysis of *S. grosvenorii*. For instance, high performance liquid chromatography (HPLC) can be used as a tool for the qualitative and quantitative analysis of chemical constituents from *S. grosvenorii*, especially for mogroside. According to the 2020 edition of Chinese Pharmacopoeia, the content of mogroside V (**16**) must be no less than 0.5% based on HPLC calibration Standard Operating Procedure (SOP). Furthermore, chromatographic separation should be conducted on a C-18 silica gel column with a mixture of acetonitrile and water at a ratio of 23:77 (V/V) used as the mobile phase, with a detection wavelength in 203 nm [10]. Zhou [79] et al. successfully established a sensitive, rapid, accurate, and reproducible HPLC-UHPLC-QTRAP^®^/MS2 method for the determination of 27 unactivated FAAs and small peptides in *S. grosvenorii*. This method can separate polar compounds within 7.5 min. Luo [80] et al. established and validated the HPLC-ESI-MS/MS system for the simultaneous quantification of 8 major sweeteners in different batches of the fruits of *S. grosvenorii* and marketed sweeteners. This method has been successfully applied to the quality control of the fruits of *S. grosvenorii* and marketed sweeteners. Systematically, target compounds in *S. grosvenorii* were screened by employing HPLC-Q-TOF-MS in combination with a screening strategy. A total of 122 compounds, including 53 flavonols and flavonol glycosides, 59 triterpenoid glycosides and 10 siraitic acid glycosides, were screened and identified from fruits, roots, stems, and leaves at 10-, 50- and 80-days of growth [12]. The HPLC fingerprint method has also been used for the quality control of *S. grosvenorii*. 11 compounds were identified on the chromatogram by comparing the retention times and UV spectra of each peak against external references, allowing effective identification and differentiation between different sources of the fruits of *S. grosvenorii* [81]. Meanwhile, the determination of grosvenorine (**74**) and kaempferitrin (**75**) was accomplished using UPLC [82]. The content of grosvenorine (**74**) and kaempferitrin (**75**) in the different parts could be determined following the order: peel (0.472 mg/g and 0.095 mg/g) > pulp (0.218 mg/g and 0.028 mg/g) > seeds. For the determination of the total flavonoid content in fruit, stem, and leaf extracts by spectrophotometry, good linearity was obtained in the range of 0.1 to 0.4 mg/mL with kaempferitrin (**75**) as a control [83].

## 5. Pharmacological Activities

*S. grosvenorii* has been used as a traditional medicine for treating asthma, tonsillitis, and sore throats. The traditional medicinal value of *S. grosvenorii* inspired extensive pharmacological research, which has brought considerable attention to its activities. In recent years, pharmacological research has found that *S. grosvenorii* possesses multi-pharmacological activities, including antitussive, expectorant, antioxidant, hypoglycemic, immunologic, hepatoprotective, and antimicrobial activities. This provide more scientific evidence for the clinical usage of *S. grosvenorii*.

### 5.1. Antitussive, Expectorant and Anti-Asthmatic Activities

Antitussive, expectorant, and anti-asthmatic activities of *S. grosvenorii* have been widely reported. Aqueous extract of *S. grosvenorii* (SGA) significantly inhibited coughs induced by concentrated ammonia or sulfur dioxide (SO_2_) in mice, as well as increasing the secretion of phenolsulfonphthalein in mice and the excretion of sputum in rats, resulting in a visible expectorant activity [84]. Sung [85] et al. investigated the anti-asthmatic activity of *S. grosvenorii* residual extract (SGRE) against ovalbumin (OVA)-induced asthma in mice. The study showed that oral administration of SGRE significantly reduced Th2 cytokines (IL-4, IL-5, and IL-13) and increased the Th1cytokine IFN-γ in the BAL fluid and supernatant of splenocyte cultures. SGRE decreased the OVA-induced increase of IL-13, TARC, MUC5AC, TNF-a, and IL-17 expression in the lung. Mogrosides could inhibit ammoniainduced cough in mice and promote mucus movement in the esophagus of frogs. According to the study, mogrosides (50, 100, and 200 mg/kg) enhanced sputum secretion in mice in a dose-dependent manner, and topical application of mogrosides enhanced ciliary cell motility in the frog respiratory tract. The results from intragastric administration showed that the anti-cough activity achieved by mogrosides positively correlated with dose, at a minimum inhibitory concentration of 80 mg/kg [86].

### 5.2. Antioxidant

Mogrosides have specific scavenging activities on hydroxyl and superoxide anion free radicals, which could reduce the occurrence of erythrocyte haemolysis, inhibit malondialdehyde production in liver mitochondria and oxidative haemolysis in rat erythrocytes, inhibit lipid peroxidation in rat liver tissues, and protect liver tissues from peroxidative damage caused by ferrous ions and hydrogen peroxide. A bioactivity experiment on *S. grosvenorii* polysaccharide (SGP) (molecular weight 1.93 × 103 kDa), has indicated [87] promising antioxidant properties in vitro, particularly in scavenging DPPH radicals. Among H_2_O_2_ oxidation-damaged PC12 cells, SGP reduced ROS and the percentage of apoptotic and necrotic cells in a dose-dependent way. Clearly, mogroside V (**16**) and 11-oxomogroside V (**43**) exhibited inhibitory activities on reactive oxygen species (O_2_^−^, H_2_O_2_ and -OH) and DNA oxidative damage. As compared with 11-oxo-mogroside V (**43**), mogroside V (**16**) is more active in the scavenging of -OH, and 11-oxo-mogroside V (**43**) is more active in the scavenging of O_2_^−^ and H_2_O_2_, 11-oxo-mogroside V (**43**) also inhibited OH-induced DNA damage [60]. Meanwhile, mogroside V (**16**) promotes the anti-oxidative stress capacity of skin fibroblasts by modulating the scavenging of free radicals with antioxidant enzymes, which could be used to prevent skin aging or lesions. In MSF treated with H_2_O_2_, mogrosides V (**16**) reduced ROS levels and MDA content, together with an increase in superoxide dismutase (SOD), glutathione peroxide (GSH-Px), and catalase (CAT) activities [47]. Furthermore, mogroside extracts (MGE) inhibited BSA glycation, as evidenced by the formation of fluorescent advanced glycation end products (AGEs) at 500 μg/ml with lower levels of protein carbonate and Ne-(carboxymethyl) lysine. MGE may be a potential anti-glycation treatment for diabetic problems by inhibiting protein glycation and sugar oxidation [6]. According to studies [88], mogroside IIIE (**41**) could decrease the levels of inflammatory cytokines and oxidative stress-related biomarkers. Furthermore, with intervention by mogroside IIIE (**41**), HG-induced apoptosis in podocytes was inhibited. Through activation of AMPK-SIRT1 signaling, mogroside IIIE (**41**) alleviates HG-induced inflammation and oxidative stress in podocytes. Ethanol extract of *S. grosvenorii* (SGE) significantly increased load swimming duration to exhaustion in mice. It could potentially suppress the reduction of HB and LD production during exercise and promote the HB synthesis and LD clearance in the recovery period, thus maintaining a high level of exercise capacity. The active ingredients could effectively promote the increase of SOD and GSH-Px activities in myocardial tissues of mice, and significantly inhibit the increase of MDA. Moreover, they can timely scavenge the excess free radicals produced during exercise during the recovery period and effectively prevent or resist lipid peroxidation in the body, which has obvious protective effects on the heart and other tissue damage caused by exercise. It has a positive effect on delaying the incidence and development of exercise fatigue [89].

### 5.3. Antidiabetic and Antihyperlipidemic Activies 

The hypoglycemic activity of cucurbitanes has been discovered, yet the underlying mechanism has not yet been discovered. Four cucurbitacins isolated from bitter melon (*Momordica charantia* L.) exhibit several beneficial biological activities against diabetes and obesity, whose hypoglycemic activity is associated with the enhanced effects of AMPK, a key pathway mediating glucose uptake and fatty acid oxidation, causing a decrease in fasting glucose and improving glucose tolerance in normal animals, diabetic animals, and humans [90]. Mogrol (**1**), 3-hydroxymogrol (**3**) and 3-hydroxy-25-dehydroxy-24-oxo-mogrol (**54**) are efficient AMPK activators in the HepG2 cell line. Both mogrol (**1**) and 3-hydroxymogrol (**3**) play a role in the hypoglycemic and anti-hyperlipid activities of this plant in vivo [25]. The MGE-fed diabetic mice induced a marked decrease in fasting plasma glucose (FPG), glucosylated serum protein (GSP), serum insulin and HOMA-IR in a dose-dependent manner, whereas insulin sensitivity, glucose, and insulin tolerance increased significantly. Lipid accumulation and adipose deposition improved and recovered to nearly normal at high doses [91]. Activation of AMPK signaling is dose-dependent.Accordingly, mRNA levels of hepatic glycogenicity and lipogenicity genes are down-regulated, whereas lipoxidation-related genes are up-regulated.

### 5.4. Anti-Inflammatory

*S. grosvenorii* residual extract (NHGR) is an inexpensive raw material. Sung [92] et al. investigated NHGR anti-allergic activity using in vivo models of atopic dermatitis (AD) characterized by mite allergens, alongside with its effects on keratocytes and immune cells. Treatment with NHGR in vitro restored the effects of pro-inflammatory cytokines on the increased filaggrin expression and reduced Dermatophagoides farinae mite antigen extract (DfE) induced phosphorylation of ERK, JNK, and p38, resulting in anti-inflammatory activity. Additionally, SGA enhanced monocyte phagocytosis in hydrocortisone injured mice and enhanced immune function [93]. Mogroside IIIE (**41**) is a compound with excellent anti-inflammatory properties, which has an improved effect on glucose metabolism, insulin resistance, and reproductive outcomes in gestational diabetes mellitus (GDM) mice. As a complication of the gestational disease, GDM clinically affects the health of mothers and infants. The treatment reduced inflammatory factor expression and alleviated GDM symptoms by enhancing AMPK activation, inhibiting HDAC4 expression, and decreasing G6Pase production [40]. Similarly, mogroside IIIE (**41**) inhibited lung fibroblast collagen production by blocking the direct differentiation of lung pericytes and resident fibroblasts induced by TGF-β or LPS. These findings suggest that mogroside IIIE (**41**) is an effective pulmonary fibrosis inhibitor. In vitro and in vivo models have been utilized to demonstrate that mogroside IIIE (**41**) significantly eliminates fibrosis in a mouse model of bleomycin-induced pulmonary fibrosis [41]. Acute lung injury (ALI) has a high mortality rate, but there is no effective treatment available. Molecular studies have shown that mogroside IIIE (**41**) increases AMPK phosphorylation while inhibiting toll-like receptor 4 (TLR4) overexpression. In addition, compound C (a pharmacological AMPK inhibitor) reversed the anti-inflammatory activity of mogroside IIIE (**41**) in LPS-induced ALI mice [42]. Mogroside IIIE (**41**) has the greatest inhibitory activity on NO (an important inflammatory factor) release from LPS-induced RAW264.7 cells, eliminating pulmonary fibrosis and demonstrating a decrease in myeloperoxidase (MPO) activity, collagen deposition and pathology scores. The expression of several fibrosis markers, such as liver fibrosis, has been significantly inhibited [94]. The anti-fibrotic activity of mogroside IIIE (**41**) may be due to downregulation of TLR4 signaling from fibroblasts as a result of its inhibiting activation by fibroblasts and deposition of ECM [95]. In the same way, TLR is the main innate immune factor activated by LPS. In the last few years, some evidence has shown that TLR4 activation binds to adaptor protein MyD88 and activates NF-κB [96]. Song [97] et al. provided evidence that mogroside V (**16**) has anti-asthmatic activity against OVA-induced asthma in mice. Effectively, mogroside V (**16**) reduced OVA-induced airway hyperresponsiveness and the number of inflammatory cells in bronchoalveolar lavage fluid (BALF). On histological examination, mogroside V (**16**) reduced IgE and IgG production and rectified the balance between Th1 and Th2 cytokines in asthmatic mice, which could be beneficial to potential protective mechanisms against OVA-induced asthma. Several lines of evidence suggest that microglia play a key role in the brain of AD patients and are critical in the pathogenesis of the disease, as activation of microglia promotes the release of pro-inflammatory factors that largely influence AD pathogenesis [98]. In a recent study, mogrosides V (**16**) has been proven to be active against LPS-induced neuroinflammation in microglia, significantly reducing the expression of pro-inflammatory proteins [99]. Simultaneously, mogroside IIE (**12**) inhibited trypsin and cathepsin B activity induced by cerulein and LPS in pancreatic islet cells AR42J and primary acinar (AP) cells in a dose- and time-dependent manner. Also, mogroside IIE (**12**) lower IL-9 levels in AP mice and reverse the inhibition of cytoplasmic calcium and the regulation of autophagy mediated by it [100]. 

### 5.5. Liver Protection

In THP-1 cells, high-purity mogroside V (**16**) suppresses reactive oxygen species generation and increases the expression of sequestosome-1 (SQSTM1, p62). Thus, mogroside could help treat obesity and non-alcoholic fatty liver disease (NAFLD) by strengthening fat metabolism and antioxidant function [101]. Mogroside V (**16**) also significantly ameliorate hepatic steatosis in mice fed a high-fat diet. Likewise, in free fatty acid (FFA) cultured LO2 cells, mogroside V (**16**) down-regulated de novo lipogenesis and up-regulated steatolysis and fatty acid oxidation, thereby reducing fat accumulation, improving hepatic steatosis induced by HFD, and alleviating the imbalance between lipid acquisition and lipid clearance [102].

### 5.6. Antibacterial and Antiviral Activities

General research has shown that flavonoids with fewer sugar groups have better antibacterial activity [68]. All the flavonoids have an inhibitory activity on Gram-positive bacteria but no inhibitory activity on Gram-negative bacteria. The MIC values of grosvenorine (**74**) and its four metabolites against Gram-positive bacteria were all less than 70 mg/mL, indicating that grosvenorine (**74**) and its four metabolites had antibacterial activity against gram-positive bacteria, showing the best antibacterial activity with kaempferol (**72**). MSSA are sensitive microorganisms. Afzelin (**79**), a flavonoid glycoside, has shown significant inhibitory activity against MSSA and MRSA, as well as Enterococcus. It is well known that flavonoids are strongly hydrophobic and readily penetrate bacterial phospholipid membranes to exert intracellular inhibitory activity [103]. In recent studies, β-amyrin (**68**), ergosterol peroxide (**87**), aloe-emodin (**82**), aloe-emodin acetate (**83**), and 4-hydroxybenzoic acid (**91**), all isolated from the leaves of *S. grosvenorii*. It has been demonstrated in vitro that they inhibit the growth activities of oral bacterial species, such as *Streptococcus mutans*, *Actinobacillus*, *Clostridium sclerotiorum*, and *Candida albicans.* Among all the active compounds, aloe-emodin (**93**) had the highest effects against all the tested bacteria and yeast, with MIC values ranging from 1.22 to 12.20 mg/mL [75]. As for this, researchers have discovered that ethanol-eluting via 50%, 70%, and 95% parts of *S. grosvenorii* has different inhibition rates against *Escherichia coli* bacterial biofilms (BBF), respectively. Therefore, the ethanol-extracted fraction of *S. grosvenorii* showed significantly better antibacterial activity than the water-extracted fraction [104]. The borderless transmission of coronavirus remains uncontrolled globally. The undetected severe acute respiratory syndrome coronavirus 2 (SARS-CoV-2) variant reduces the therapeutic efficacy of vaccines against coronavirus disease 2019 (COVID-19). Clinical observations suggest that tumor cases are highly infected with coronavirus, possibly due to immunologic injury, causing a higher COVID-19-related death toll [105,106]. Based on network pharmacology analysis, the current study identified 24 candidate targets and 10 core targets for mogroside V (**16**) for treating COVID-19 [107]. Mogroside V may treat COVID-19 by targeting JUN, IL2, HSP90AA1, AR, PRKCB, VEGF, TLR9, TLR7, STAT3, and PRKCA. Further analysis using molecular docking suggested potent binding scores between mogroside V (**16**) and the core target protein in COVID-19. VEGF, a core protein docked well with mogroside V (**16**), has been identified as a pharmacological target in mogroside V (**16**) treatment of COVID-19. Clinically, VEGF may induce vascular impairment in gastrointestinal cells infected with SARSCoV-2 [108]. 

### 5.7. Miscellaneous Activities

Mogroside V (**16**) has comprehensive anti-cancer activities. The natural sweet compound mogroside V (**16**) could inhibit the proliferation and survival of pancreatic cancer cells by focusing on multiple biological targets. In both in vitro and in vivo models of pancreatic cancer, mogroside V (**16**) has tumor growth inhibitory activity by promoting apoptosis and cell cycle arrest in pancreatic cancer cells (PANC-1 cells), possibly through modulation of the STAT3 signalling pathway, promoting cell proliferation (CCND1, CCNE1 and CDK2), while also upregulating cell cycle inhibitors (CDKN1A and CDKN1B) [109]. Mogroside V (**16**) inhibited hypoglycemic-induced lung cancer cell migration and invasion by reversing EMT and disrupting the cytoskeleton. In lung cancer cells cultured under hypoglycemic conditions, the metastatic efficiency of mogroside V (**16**) was compared with normoglycemia, which reversed hyperglycemia-induced invasion and migration by upregulating E-Cadherin expression and downregulating N-Cadherin, Vimentin, and Snail expression. Meanwhile, the expression levels of Rho A, Rac1, Cdc42 and p-PAK1 protein were decreased in a dose-dependent manner [110]. After injection of Aβ1-42, mice showed a distinct increase in escape latency and a distinct decrease in the number of times they crossed the target and time spent in the target quadrant. Mogrol (**1**) could significantly alleviate memory impairment caused by Aβ1-42, inhibit microglia overactivation and prevent hippocampus apoptosis, downregulate the high expression of IL1β, IL-6, NF-kB p65, TNF-a induced by Aβ1–42, reduce the neuroinflammation [5]. Likewise, this plant plays a role in anti-aging. Rats fed with *S. grosvenorii* showed a slower ageing process. The experimental group maintained a stationary LSK state, decreased ROS level, augmented hematopoietic stem cells, and reduced the number of β-gal positive cells associated with aging, thus lowering the expression of aging-related proteins and slowing the aging process in rats [111]. 

### 5.8. Summary of Pharmacologic Activities

In conclusion, *S. grosvenorii* has a wide range of pharmacological activities (Table 4). Modern pharmaceutical research mainly focuses on extracts and chemical components, indicating the prospects of *S. grosvenorii* in the treatment of such diseases.

## 6. Toxicological Profile: Safety and Adverse Activities

In order to ensure that *S. grosvenorii* is commercially available for use as a medicine and food, its acute toxicity, chronic toxicity, and genotoxicity need to be determined.

Researchers used the AESG (mogrosides content more than 80%) to treat Kunming mice with a single gavage and observed toxic reactions at the highest dose of 24 g/kg for 7 days. No behavioral manifestations or body weight abnormalities were observed, and no animal death was observed; at the highest dose of 120 g/kg for 14 days. None of the experimental animals in each dose group died [112]. These results proved that mogrosides are essentially nontoxic substances. Researchers conducted repeated dose toxicity experiments on rats for 13 weeks with diets containing 0.00%, 0.04%, 0.20%, 1.00%, and 5.00% of *S. grosvenorii* extract (containing mogroside V). The results showed that rats in each dose group had no mortality within 13 weeks and no changes in body weight, general behavior, blood biochemical, and histopathology parameters. Apart from this, the extract had no observed adverse effect level (NOAEL) of more than 5% (2520 mg/kg/day for males and 3200 mg/kg/day for females) in Wistar Hannover rats [113]. The researchers performed sperm aberration tests after gavage treatment of 2 batches of mice with the aqueous extracts. The experimental results showed no statistical difference in the rates of cell micronuclei and sperm aberrations in each dose group compared to the negative control group (*p* > 0.05) [114]. However, other research has shown that rates of cell micronuclei and sperm aberrations were higher in the high dose group (5 g/kg) than in the negative control group (*p* < 0.05), while the remaining dose groups were not statistically different from the negative control group [115]. Therefore, whether mogrosides are toxic needs to be further explored, and these results also suggest that the dose of natural active substance used should be focused on, which still carries a risk of genotoxicity if the dose is too high.

## 7. Conclusions

The review summarized the main types of secondary metabolites in *S. grosvenorii*, including triterpenoids, flavonoids, lignans, alkaloids, fatty acids, and others. Most of cucurbitanes triterpenoids showed strong inhibitory activity on inflammation. *S. grosvenorii* has been reported to be rich in mogrosides as one of its characteristic constituents, which have become a hot topic of research in recent years. Few systematic studies have reported mogrosides due to their high impact biological activities. These findings may be great importantance, providing a foundation for finding more mogrosides and cucurbitanes. As a famous traditional medicine with a long history, previous studies of *S. grosvenorii* have focused on the composition and its traditional pharmacological activity. *S. grosvenorii* has a wide range of biological activities, including anti-tussive, expectorant, anti-asthmatic activities, antioxidants, hypoglycemic, immune and anti-inflammatory, and liver protection. Further research into these pharmacological activities is still worthwhile. Moreover, clinical studies should be conducted to comprehensively assess the therapeutic effects, adverse effects, and toxicity of *S. grosvenorii*. Among the several active compounds found in *S. grosvenorii*, mogrosides are considered to be the main active constituents and undertake much of the pharmacological activities, but the interactions of mogrosides with other constituents cannot be excluded. Flavonoids, sterols, phenolic acids and other compounds enhance pharmacological activities to a certain extent. Besides mogroside monomers, the biological activities of others and the interaction between mogrosides and others are future challenges. In recent years, phytochemistry and pharmacological studies of *S. grosvenorii* have received considerable attention, yet market demands, and quality control should also receive high priority. The information on the action mechanisms for this plant will aid its future development as a new health food or therapeutic agent. *S. grosvenorii* is one of the first herbs used for both medicinal and food purposes in China. There are many foods or drinks on the market that are made from the fruits of *S. grosvenorii*, which have certain health functions and can improve living standards. Hunan Huacheng Biotech, Inc. has been dedicated to bringing healthy sweeteners to everyone. In summary, the medicinal and edible values of *S. grosvenorii* are of inestimable value for further research in the future. Although the specific mechanisms of action need to be further explored, its active constituents show great potential for medical and lifestyle applications, and further studies will provide new ideas for solving human health problems.

## Figures and Tables

**Figure 1 molecules-27-06618-f001:**
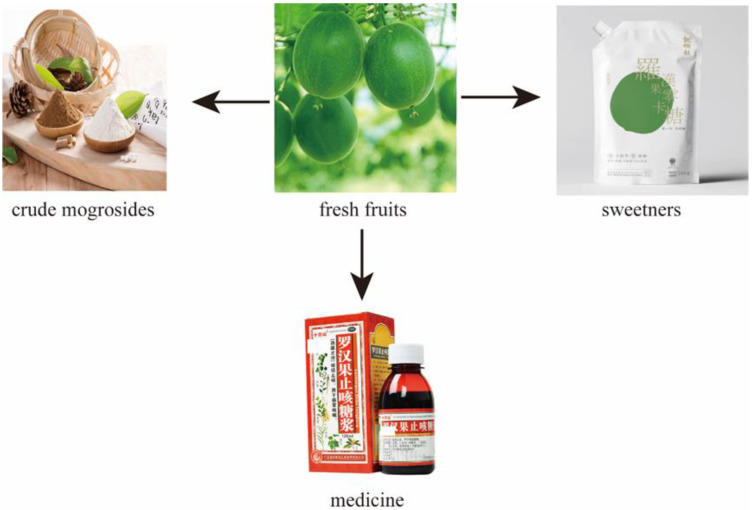
Applications of *Siraitia grosvenorii*.

**Figure 2 molecules-27-06618-f002:**
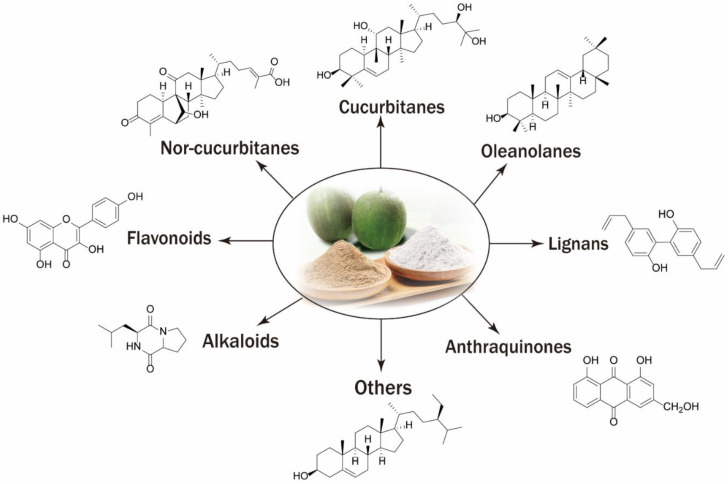
Main types of secondary metabolites in *Siraitia grosvenorii*.

**Figure 3 molecules-27-06618-f003:**
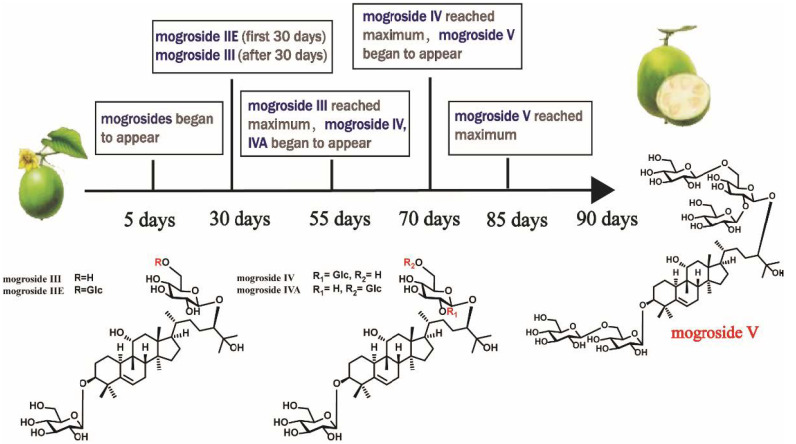
Structural changes in mogroside during different growth periods.

**Figure 4 molecules-27-06618-f004:**
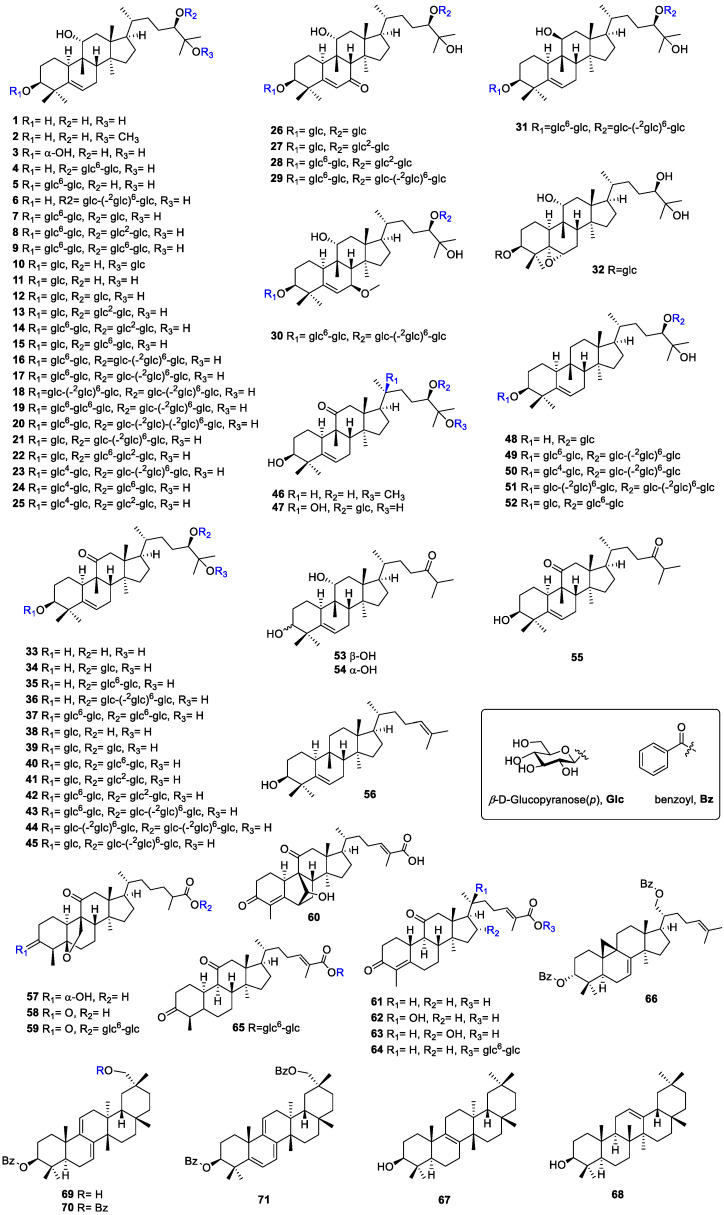
The structures of triterpenoids from *S. grosvenorii*.

**Figure 5 molecules-27-06618-f005:**
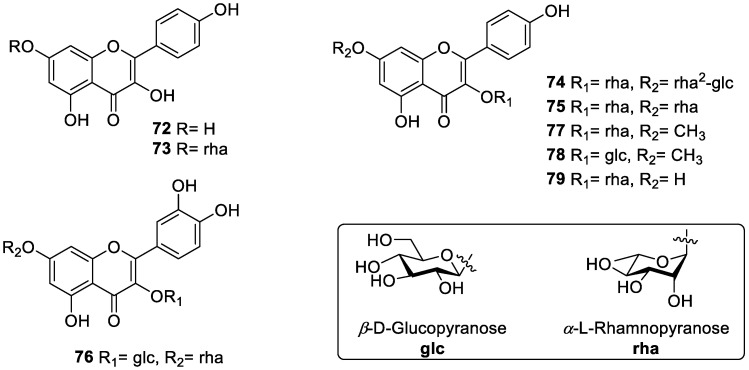
The structures of flavonoids from *S. grosvenorii*.

**Figure 6 molecules-27-06618-f006:**
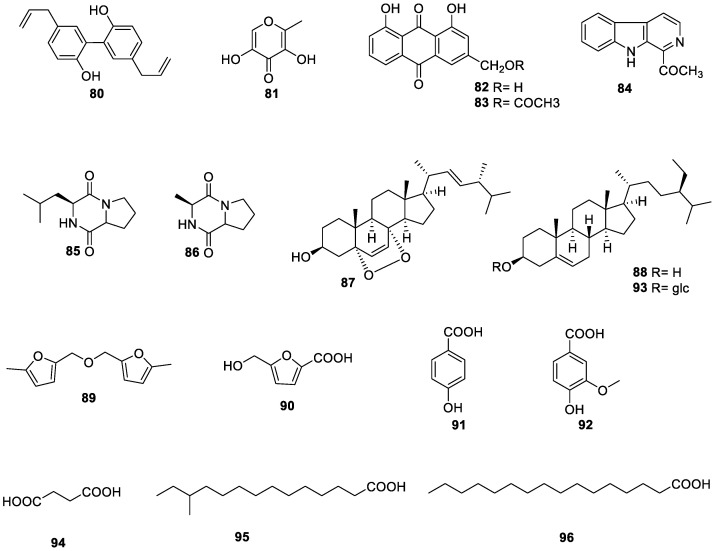
The structures of miscellaneous compounds from *S. grosvenorii*.

**Table 1 molecules-27-06618-t001:** The main cucurbitanes from *S. grosvenorii*.

No.	Compound Name	Pharmacological Activities	Origins	Ref.
**1**	mogrol	neuroprotective	fruit	[16,34]
**2**	25-methoxymogrol	antidiabetic, anti-hyperlipid	fruit	[25]
**3**	3α-hydroxymogrol	antidiabetic, anti-hyperlipid	fruit	[25]
**4**	mogroside IIA1	antiviral	fruit	[35]
**5**	mogroside IIA2	antidiabetic, antioxidant	fruit	[6,36]
**6**	mogroside IIIA1	antidiabetic	fruit	[36]
**7**	mogroside IIIA2	antiviral	fruit	[16,35]
**8**	mogroside IV	antidiabetic	fruit	[16,37]
**9**	mogroside IVA	-	fruit	[16,17]
**10**	mogroside IIB	-	fruit	[35]
**11**	mogroside IE1	-	fruit	[8,16]
**12**	mogroside IIE	antidiabetic, antioxidant	fruit	[8,38,39]
**13**	mogroside IIIE	anti-inflammatory, antidiabetic, anti-fibrotic	fruit	[16,40,41,42]
**14**	mogroside IVE	anti-fibrotic, anti-tumor	fruit	[43,44,45]
**15**	mogroside III	-	fruit	[8,17,46]
**16**	mogroside V	anti-inflammatory, antioxidant	fruit	[8,16,47,48,49]
**17**	mogroside VA1	-	fruit	[50]
**18**	mogroside VI	liver protective	fruit	[16,51]
**19**	mogroside VIA	-	fruit	[52]
**20**	mogroside VIB	-	fruit	[52]
**21**	siamenoside I	-	fruit	[8,45,49,53]
**22**	grosmomoside I	-	fruit	[54]
**23**	isomogroside V	-	fruit	[45,55]
**24**	isomogroside IVa	-	fruit	[36]
**25**	isomogroside IVe	-	fruit	[36]
**26**	7-oxo-mogroside IIE	-	fruit	[56]
**27**	7-oxo-mogroside IIIE	-	fruit	[52]
**28**	7-oxo-mogroside IV	-	fruit	[52]
**29**	7-oxo-mogroside V	-	fruit	[56]
**30**	7β-methoxy-mogroside V	-	fruit	[11]
**31**	11-epimogroside V	-	fruit	[8,45,49,53]
**32**	5α,6α-epoxymogroside IE1	-	fruit	[26]
**33**	11-oxo-mogrol	liver protective	fruit	[57,58]
**34**	11-oxo-mogroside IA1	-	fruit	[17,58]
**35**	11-oxomogroside IIA1	-	fruit	[26]
**36**	11-oxo-mogroside IIIA1	-	fruit	[11]
**37**	11-oxo-mogroside IVA	-	fruit	[45]
**38**	11-oxo-mogroside IE1	-	fruit	[16,58]
**39**	11-oxo-mogroside IIE	-	fruit	[17]
**40**	11-oxo-mogroside III	-	fruit	[59]
**41**	11-oxo-mogroside IIIE	-	fruit	[52]
**42**	11-oxo-mogroside IV	-	fruit	[52]
**43**	11-oxo-mogroside V	antioxidant, anti-tumor	fruit	[45,60,61]
**44**	11-oxo-mogroside VI	antidiabetic	fruit	[36]
**45**	11-oxo-siamenoside I	antidiabetic	fruit	[36]
**46**	25-methoxy-11-oxomogrol	antidiabetic, anti-lipidemic	fruit	[25]
**47**	20-hydroxy-11-oxomogroside I A1	-	fruit	[17]
**48**	mogroside IA (mogroside, A1)	-	fruit	[16]
**49**	11-deoxymogroside V	antidiabetic	fruit	[36,62]
**50**	11-deoxyisomogroside V	-	fruit	[62]
**51**	11-deoxymogroside VI	-	fruit	[62]
**52**	11-deoxymogroside III	-	fruit	[35,59]
**53**	25-dehydroxy-24-oxomogrol	antidiabetic, anti-lipidemic	fruit	[25]
**54**	3-hydroxy-25-dehydroxy-24-oxomogrol	antidiabetic, anti-lipidemic	fruit	[25]
**55**	bryogenin	antidiabetic, anti-lipidemic	fruit	[25]
**56**	10α-cucurbitadienol	-	seed oil	[26]

**Table 2 molecules-27-06618-t002:** Flavonoids isolated from *S. grosvenorii*.

No.	Compounds Name	Pharmacological Activities	Origins	Ref.
**72**	kaempferol	antioxidant, anti-inflammatory	flower, leaf	[65,69,70]
**73**	kaempferol-7-O-α-L-rhamnopyranoside	antioxidant	flower, leaf, fruit	[70,71]
**74**	grosvenorine	antioxidant	flower, leaf, fruit	[63,70,71]
**75**	kaempferitrin	antioxidant, anti-inflammatory, anti-convulsant	leaf, fruit	[63,65,71,72]
**76**	quercetin 3-O-β-D-glucopyranosyl 7-O-α-L-rhamnopyranoside	-	leaf, fruit	[63,65,71]
**77**	7-methoxy-kaempferol 3-O-α-L-rhamnopyranoside	antioxidant	flower	[70]
**78**	7-methoxy-kaempferol 3-O-β-D-glucopyranoside	antioxidant	flower	[70]
**79**	afzelin	anti-tumor, anti-inflammatory	fruit	[68,73,74]

**Table 3 molecules-27-06618-t003:** Other compounds isolated from *S. grosvenorii*.

No.	Compounds Name	Pharmacological Activities	Origins	Ref.
**80**	magnolol	-	fruit	[27]
**81**	5-hydroxymaltol	-	fruit	[66]
**82**	aloe emodin	antibacterial	leaf	[75]
**83**	aloe-emodin acetate	antibacterial	leaf	[75]
**84**	1-acetyl-β-carboline	-	fruit	[66]
**85**	cyclo-(leu-pro)	-	fruit	[66]
**86**	cyclo-(ala-pro)	-	fruit	[66]
**87**	ergosterol peroxide	antibacterial	leaf	[75]
**88**	β-sitosterol	-	fruit	[66]
**89**	5,5′-oxydimethylene-bis-(2-furfural)	-	fruit	[27]
**90**	5-(hydroxymethyl)-2-furancarboxylic acid	-	fruit	[27]
**91**	4-hydroxybenzoic acid	antibacterial	leaf	[75]
**92**	vanillic acid	-	fruit	[66]
**93**	daucosterol	-	leaf	[75]
**94**	succinic acid	-	fruit	[27]
**95**	12-methyltetradecanoic acid	-	leaf	[75]
**96**	n-hexadecanoic acid	-	leaf	[75]

**Table 4 molecules-27-06618-t004:** Pharmacological activities of *S. grosvenorii*.

Activities	Detail	Extracts/Compounds	Concentration/Dose	In Vivo/In Vitro	Ref.
Antitussive, expectorant and anti-asthmatic activities	increased the secretion of phenolsulfonphthalein in mice and the excretion of sputum in rats	SGA	4000 and 8000 mg/kg	in vivo	[84]
	reduced Th2 cytokines (IL-4, IL-5, and IL-13) and increased the Th1cytokine IFN-γ	SGRE	200 mg/kg	in vivo	[85]
	enhanced sputum secretion in mice and ciliary cell motility in the frog respiratory tract	mogrosides	50, 100, and 200 mg/kg	in vivo	[86]
Antioxidant	decrease ROS in oxide injury PC12 cells and decrease apoptotic and necrotic cells	SGP	0.5, 1.0, 1.5, 2.0 mg/mL	in vitro	[87]
	reduced ROS levels and MDA content, increase SOD, GSH-Px and CAT activities	mogroside V	30, 60, and 90 µg/mL	in vitro	[47]
	scavenging of -OH	mogroside V	PC12 cells EC50 = 48.44 μg/mL	in vitro	[60]
	scavenging of O_2_ and H_2_O_2_, inhibited -OH induced DNA damage	11-oxomogroside V	PC12 cells EC50 = 4.79 μg/mL, EC50 = 16.52 μg/mL, EC50 = 3.09 μg/mL	in vitro	[60]
	inhibited BSA glycation	MGE	500 μg/mL	in vitro	[6]
	decrease the levels of inflammatory cytokines and oxidative stress-related biomarkers	mogroside IIIE	MPC-5 cells 1, 10, and 50 μM	in vitro	[88]
	inhibited the reduction of HB and LD and inhibited the increase of MDA, promoted the synthesis of HB and the clearance of LD	SGE	1500 mg/kg	in vivo	[89]
Hypoglycemic	increase AMPK phosphorylation	mogrol	HepG2 cell line 1, 10, and 20 μM	in vivo	[25]
	increase AMPK phosphorylation	3-hydroxymogrol	HepG2 cell line 1, 10, and 20 μM	in vivo	[25]
	increase AMPK phosphorylation	3-hydroxy-25-dehydroxy-24-oxo-mogrol	HepG2 cell line 4 μM	in vivo	[25]
	downregulated mRNA levels of hepatic gluconeogenic and lipogenic genes, upregulated fat oxidation-associated genes	MGE	300 mg/kg	in vivo	[91]
Immunology and anti-inflammatory	increased filaggrin expression and reduced DfE induced phosphorylation of ERK, JNK and p38	NHGR	200 or 400 mg/kg	in vivo	[92]
	enhanced monocyte phagocytosis in hydrocortisone injured	SGA	2500 and 5000 mg/kg	in vivo	[93]
	decreased expression of IL-1b, IL-6, and TNF-a	mogroside IIIE	GDM model 20.0 mg/kg	in vivo	[40]
	inhibited LPS-induced inflammatory	mogroside IIIE	RAW264.7 cells 10 μM	in vitro	[40]
	inhibited LPS-induced inflammatory	mogroside IIIE	RAW264.7 cells 1, 10 or 50 μM	in vitro	[42]
	reduced the OVA-induced activation of NF-κB	mogroside V	2, 5, and 10 mg/kg	in vivo	[97]
	inhibited LPS-induced inflammatory	mogroside V	BV-2 cells 6.25, 12.5 and 25 μM	in vitro	[99]
	inhibited the IL-9/IL-9R/calcium overload/cathepsin B activation/trypsinogen activation pathway	mogroside IIE	AR42J cells 5, 10 and 20 μM	in vitro	[100]
Liver protection	inhibited reactive oxygen species production and upregulated sequestosome-1 (SQSTM1, p62) expression	mogroside V	200, 400, and 800 mg/kg	in vivo	[101]
	activated AMPK ameliorates HFD-induced hepatic steatosis	mogroside V	LO2 cells 15, 30, 60, and 120 μM	in vitro	[102]
Antibacterial and anti-viral	inhibitory effects against gram-positive bacteria	grosvenorine	MIC less than 70 μg/mL	in vitro	[68]
	inhibitory effects against gram-positive bacteria	kaempferitrin	MIC less than 70 μg/mL	in vitro	[68]
	inhibitory effects against gram-positive bacteria and MSSA	kaempferol	MIC less than 70 μg/mL	in vitro	[68]
	inhibitory effects against MSSA and MRSA	afzelin	MIC less than 70 μg/mL	in vitro	[68]
	inhibitory effects against Streptococcus mutans, Actinobacillus actinobacillus, Clostridium sclerotiorum, and Candida albicans	β-amyrin	MIC = 48.80, >100, 48.40, and >100 μg/mL	in vitro	[75]
	inhibitory effects against Streptococcus mutans, Actinobacillus actinobacillus, Clostridium sclerotiorum, and Candida albicans	ergosterol peroxide	MIC = 4.88, 48.80, 48.80, and 12.20 μg/mL	in vitro	[75]
	inhibitory effects against Streptococcus mutans, Actinobacillus actinobacillus, Clostridium sclerotiorum, and Candida albicans	aloe-emodin	MIC = 1.22, 6.10, 12.20, and 6.10 μg/mL	in vitro	[75]
	inhibitory effects against Streptococcus mutans, Actinobacillus actinobacillus, Clostridium sclerotiorum, and Candida albicans	aloe-emodin acetate	MIC = 6.10, 12.20, >100, and 6.10 μg/mL	in vitro	[75]
	inhibitory effects against Streptococcus mutans, Actinobacillus actinobacillus, Clostridium sclerotiorum, and Candida albicans	4-hydroxybenzoic acid	MIC = 12.20, >100, 12.20, and 12.20 μg/mL	in vitro	[75]
	inhibitory effects against Escherichia coli bacterial biofilms	different ethanol-eluting parts of *S. grosvenorii*	MIC = 55.58, 78.32, and 87.62%	in vitro	[103]
	regulation of VEGF	mogroside V	-	in vitro	[107]
Miscellaneous activities	inhibited expression of the STAT3 pathway	mogroside V	PANC-1 cells 10, 100, and 250 μM	in vivo and in vitro	[109]
	inhibited expression levels of Rho A, Rac1, Cdc42 and p-PAK1	mogroside V	A549 and H1299 cells 0–50 μM	in vitro	[110]
	inhibited expression of IL1β, IL-6, NF-κB p65, TNF-a induced by Aβ1–42	mogrol	20, 40, 80 mg/kg	in vivo	[5]
	decrease the intercellular levels of ROS	extract of *S. grosvenorii*	200 mg/kg	in vivo	[111]

## Data Availability

Not applicable.

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
