# Peer review of "A Review of the Phytochemistry and Pharmacology of the Fruit of Siraitia grosvenorii (Swingle): A Traditional Chinese Medicinal Food"

_molecules, 2022, doi:10.3390/molecules27196618_

Round 1

Reviewer 1 Report

Dear Authors

Thank you for giving me the chance to look at your work. The review is valuable and interesting.

Title : Suggest the change of title to : A review of the phytochemistry and pharmacology of the fruit of Siraitia grosvenorii :a traditional chinese medicinal food . There are some grammars and punctation mistakes I have corrected some and highlighted. The language could be better revised by a native speaker.

Introduction:

Add a reference for distribution of the genus. Also add a short description for the fruits after vine. The search conducted using Google scholar etc. and the years covered.

Line

Comment

 45

Mogrosides, a series of responsible for sweetness and some

61

Also the fruits

98

Have  mogrol

267

 Types of flavonoids are few unlimited

287

Miscellanoues 

357

Scientific evidence

413

Antidiabetic and antihyperlipidemic  activity

431

Antiinflammatory activity

433

In vivo italics

435

In vitro italics

491

Antibacterial and antiviral activity

505

Streptococcus …. Capital and italics                                                                             

Author Response

Dear reviewer, I will answer your questions point-by-point as follows.

Point 1: Title: Suggest the change of title to: A review of the phytochemistry and pharmacology of the fruit of Siraitia grosvenorii :a traditional Chinese medicinal food. There are some grammars and punctation mistakes I have corrected some and highlighted. The language could be better revised by a native speaker.

Response 1: Please kindly see the title, I have been updated.

Point 2: Add a reference for distribution of the genus. Also add a short description for the fruits after vine. The search conducted using Google scholar etc. and the years covered.

Response 2: The reference has been updated, please kindly see line 34 in the revised manuscript. A short description has been added, please kindly see lines 39-42 in the revised manuscript.

Minor points (partly indicated, for not all)

  1. (L16) Southern

Response: The “southern” was corrected as “Southern”. Please kindly see line 35 in the revised manuscript.

  1. (L21) previous

Response: The “Previous” was corrected as “previous”. Please kindly see line 21 in the revised manuscript.

  1. (L27) of its biological functions

Response: The “for its biological in functions” was corrected as “of its biological functions”. Please kindly see line 27 in the revised manuscript.

  1. (L36) grosvenorii

Response: The “S. grosvenorii” was corrected as “Siraitia. grosvenorii”. Please kindly see line 36 in the revised manuscript.

  1. (L42) healthy

Response: The “health” was corrected as “healthy”. Please kindly see line 42 in the revised manuscript.

  1. (L42) pharmacological activities

Response: The “pharmacological activities” was added. Please kindly see line 42 in the revised manuscript.

  1. (L44) other activities

Response: The “others” was corrected “other activities”. Please kindly see line 44 in the revised manuscript.

  1. (L52) Mogrosides, a series of responsible for sweetness and some

Response: The “Mogroside, a series of triterpenoid glycosides” was corrected “Mogrosides, a series of responsible for sweetness and some”. Please kindly see line 52 in the revised manuscript.

  1. (L86) Also, the fruits of grosvenorii

Response: The “The fruits of S. grosvenorii” was corrected “Also, the fruits of S. grosvenorii”. Please kindly see line 86 in the revised manuscript.

  1. (L98) sore throat

Response: The “sore-throat” was corrected “sore throat”. Please kindly see line 98 in the revised manuscript.

  1. (L129-130) All these compounds have mogrol, [10-cucurbit-5-ene-3,11,24R,25-tetraol], attached to different numbers of glucose units.

Response: The “All of these compounds have the mogrol, an aglycone of mogrosides, [10-cucurbit-5-ene-3,11,24R,25-tetraol], that attached to different numbers of glucose units.” was corrected “All these compounds have mogrol, [10-cucurbit-5-ene-3,11,24R,25-tetraol], attached to different numbers of glucose units.”. Please kindly see lines 129-130 in the revised manuscript.

  1. (L207) were very few

Response: The “were very limited” was corrected “were very few”. Please kindly see line 207 in the revised manuscript.

  1. (L227) Miscellaneous componds

Response: The “Other componds” was corrected “Miscellaneous componds”. Please kindly see line 227 in the revised manuscript.

  1. (L287) scientific evidence

Response: The “scientific interpretation” was corrected “scientific evidence”. Please kindly see line 287 in the revised manuscript.

  1. (L299) ammonia induced

Response: The “ammonia-induced” was corrected “ammonia induced”. Please kindly see line 299 in the revised manuscript.

  1. (L345) Antidiabetic and antihyperlipidemic activies

Response: The “Hypoglycemic” was corrected “Antidiabetic and antihyperlipidemic activies”. Please kindly see line 345 in the revised manuscript.

  1. (L363) Anti-inflammatory

Response: The “Immunology and anti-inflammatory” was corrected “Anti-inflammatory”. Please kindly see line 363 in the revised manuscript.

  1. (L365 and 367) in vivo/vitro italics

Response: The “in vivo/vitro” was corrected italics. Please kindly see lines 365 and 367 in the revised manuscript.

  1. (L424) Antibacterial and antiviral activities

Response: The “Antibacterial and anti-viral” was corrected “Antibacterial and antiviral activities”. Please kindly see line 424 in the revised manuscript.

  1. (L427 and 428) Gram-negative/positive

Response: The “gram-negative/positive” was corrected “Gram-negative/positive”. Please kindly see lines 408 and 409 in the revised manuscript.

  1. (L438-439) Streptococcus …. Capital and italics

Response: The “Streptococcus mutans, Actinobacillus, Clostridium sclerotiorum, and Candida albicans.” was corrected. Please kindly see lines 438-439 in the revised manuscript.

Reviewer 2 Report

The manuscript entitled '' Phytochemistry and pharmacology of the fruit of Siraitia grosvenorii: a review of traditional Chinese medicine with same origins of medicinal and food'' is interesting, well-written, and organized. However, the authors need to make some changes

·       Siraitia grosvenorii should be written with its authority name in the title

·       Regarding active constituents, please refer to the characteristic marker compounds of this family and their existence in S. grosvenorii.

·       Please, check the abbreviations throughout the manuscript. It would be best to introduce the abbreviation when the whole word appears the first time in the text and then use only the abbreviation.

·       The Figures should be self-explanatory. Therefore, figures 3,4,5 should include a legend that links the numbers and structure names. Figure 2 has a too small character size; therefore hard to read axis labels. Please improve and revise the number of figures in the manuscript body.

·       Please align the content in Tables 1-3 left or center, not align right.    

·       I wonder if there are approved market or pharmaceutical preparations of S. grosvenorii fruit. The author should discuss this point. They mentioned it as sweeteners.

·      Grammatical, alignment and typographical errors are noted in the manuscript, and they should be thoroughly checked and corrected throughout the manuscript.

·       In the conclusion of your review, it should contain more analysis of the data collected and not a compilation of the data without criticized comments.

Author Response

Dear reviewer, I will answer your questions point-by-point as follows.

Point 1: Siraitia grosvenorii should be written with its authority name in the title.

Response 1: The title with authority name is too lengthy, so the authority name of Siraitia grosvenorii has been filled in when it first appears in the abstract and introduction. Please kindly see the abstrct (line 15) and introductin (line 36).

Point 2: Regarding active constituents, please refer to the characteristic marker compounds of this family and their existence in S. grosvenorii.

Response 2: The active constituents have already been demonstrated in the phytochemistry and pharmacology parts, and I have made some modifications based on this. please kindly see the revision.

Point 3: Please, check the abbreviations throughout the manuscript. It would be best to introduce the abbreviation when the whole word appears the first time in the text and then use only the abbreviation.

Response 3: All abbreviations have been carefully checked and revised. Please kindly see line 290, line 327, lines 350, line 370, lines 374-375, lines 398, line 410, line 411 and line 492 in revision.

Point 4: The Figures should be self-explanatory. Therefore, figures 3,4,5 should include a legend that links the numbers and structure names. Figure 2 has a too small character size; therefore hard to read axis labels. Please improve and revise the number of figures in the manuscript body.

Response 4: All figures have been carefully checked and revised. The links between numbers in figures and structure names in tables have been described in the text (Table1 and Figue 4; Table 2 and Figure 5; Table 3 and Figue 6), please kindly see lines 196-198, lines 221-223, and lines 231-233 in revision. The Figure 2 has been enlarged and resized to fit, I have been added a figure to display “The mogrosides content in fruits during different growth periods.”. Please kindly see the revision.

Point 5: Please align the content in Tables 1-3 left or center, not align right.

Response 5: All tables have been carefully checked and revised. Please kindly see the revision.

Point 6: I wonder if there are approved market or pharmaceutical preparations of S. grosvenorii fruit. The author should discuss this point. They mentioned it as sweeteners.

Response 6: The discussion about the approved market or pharmaceutical preparations of S. grosvenorii fruit were updated. Please kindly see lines 63-76.

Point 7: Grammatical, alignment and typographical errors are noted in the manuscript, and they should be thoroughly checked and corrected throughout the manuscript.

Response 7: The grammatical, alignment and typographical errors were checked and corrected. Please kindly see the revision.

Point 8: In the conclusion of your review, it should contain more analysis of the data collected and not a compilation of the data without criticized comments.

Response 8: This review mainly focuses on the phytochemistry and pharmacology of S. grosvenorii. The side effects of this plant have not been fully performed so far. Additionally, we have critically discussed the relative prospective trends on this in conclusion part. Please kindly see lines 543-546 in revised manuscript.

Round 2

Reviewer 2 Report

The authors thoroughly revised the manuscript. So I recommend publishing it